# In Vitro Generation of Haploid Germ Cells from Human XY and XXY Immature Testes in a 3D Organoid System

**DOI:** 10.3390/bioengineering11070677

**Published:** 2024-07-03

**Authors:** Guillermo Galdon, Nima Pourhabibi Zarandi, Nicholas A. Deebel, Sue Zhang, Olivia Cornett, Dmitry Lyalin, Mark J. Pettenati, YanHe Lue, Christina Wang, Ronald Swerdloff, Thomas D. Shupe, Colin Bishop, Kimberly Stogner, Stanley J. Kogan, Stuart Howards, Anthony Atala, Hooman Sadri-Ardekani

**Affiliations:** 1Wake Forest Institute for Regenerative Medicine (WFIRM), Wake Forest School of Medicine, Winston-Salem, NC 27101, USA; 2Facultad de Medicina, Universidad de Barcelona, 08036 Barcelona, Spain; 3Department of Internal Medicine, University of Pittsburgh Medical Center, Harrisburg, PA 17101, USA; 4Department of Urology, Wake Forest School of Medicine, Winston-Salem, NC 27157, USA; 5Department of Biomedical Engineering, Boston University, Boston, MA 02215, USA; 6Department of Pathology, Wake Forest School of Medicine, Winston-Salem, NC 27157, USA; 7Department of Pathology, Molecular Diagnostics Division, Virginia Commonwealth University, Richmond, VA 23284, USA; 8Division of Endocrinology, Department of Medicine, The Lundquist Institute, Harbor-University of California Los Angeles (UCLA) Medical Center, Los Angeles, CA 90502, USA

**Keywords:** spermatogonia, spermatogonia stem cells, in vitro, spermatogenesis 3D culture, organoid, Klinefelter syndrome, male infertility, fertility preservation, cancer survivors

## Abstract

Increasing survival rates of children following cancer treatment have resulted in a significant population of adult survivors with the common side effect of infertility. Additionally, the availability of genetic testing has identified Klinefelter syndrome (classic 47,XXY) as the cause of future male infertility for a significant number of prepubertal patients. This study explores new spermatogonia stem cell (SSC)-based fertility therapies to meet the needs of these patients. Testicular cells were isolated from cryopreserved human testes tissue stored from XY and XXY prepubertal patients and propagated in a two-dimensional culture. Cells were then incorporated into a 3D human testicular organoid (HTO) system. During a 3-week culture period, HTOs maintained their structure, viability, and metabolic activity. Cell-specific PCR and flow cytometry markers identified undifferentiated spermatogonia, Sertoli, Leydig, and peritubular cells within the HTOs. Testosterone was produced by the HTOs both with and without hCG stimulation. Upregulation of postmeiotic germ cell markers was detected after 23 days in culture. Fluorescence in situ hybridization (FISH) of chromosomes X, Y, and 18 identified haploid cells in the in vitro differentiated HTOs. Thus, 3D HTOs were successfully generated from isolated immature human testicular cells from both euploid (XY) and Klinefelter (XXY) patients, supporting androgen production and germ cell differentiation in vitro.

## 1. Introduction

The mammalian testis is a complex multicellular organ that can be divided into two main compartments, each playing a unique role. In the seminiferous tubules, Sertoli cells provide a supportive environment for germ cell differentiation from spermatogonia to haploid gametes [1,2]. Leydig cells within the interstitial tissue surrounding the tubules produce testosterone through steroidogenesis [3,4]. High testosterone concentration adjacent to the seminiferous tubules is essential for the completion for spermatogenesis. Maintaining these two functions requires a complex communication network among the cells with this microenvironment [5].

Genetic, epigenetic, and environmental factors can disrupt this communication network, contributing to a rise in the rate of male infertility [6,7]. Approximately 8–12% of couples trying to conceive suffer from infertility, with half of the underlying complications due to male factors [8]. Technical advancements over the past decades, such as microsurgical testicular sperm extraction (m-TESE), intracytoplasmic sperm injection (ICSI), and more recently, experimental round spermatid injection (ROSI), have provided therapeutic options for infertile men attempting to father children [9,10]. Nevertheless, these strategies all require functional differentiated haploid cells. 

Recent advances in cancer treatments have significantly increased the prevalence of male cancer survivors. Unfortunately, both cancer and cancer treatments, such as chemotherapeutic agents and radiotherapy, have the potential to damage the testis, frequently leading to male infertility [11,12,13]. Sperm cryostorage is now a clinical option for adult patients undergoing gonadotoxic treatments. However, this option is unavailable for prepubertal patients without fully established spermatogenesis. 

Similarly, Klinefelter syndrome (KS) is a chromosomal disease characterized by male phenotype and X chromosome aneuploidy, with 47,XXY being the most frequent karyotype [14,15]. Although widely underdiagnosed, KS affects 1 in every 600–1000 males and is currently considered the most common genetic cause of male infertility [16,17,18]. The most prominent clinical feature of KS is severe testicular fibrosis that accelerates germ cell loss at the onset of puberty. This causes azoospermia in most KS patients by the time they reach adulthood [19,20]. Due to the lack of available sperm, previously mentioned fertility treatments like mTESE and ICSI have only been partially successful in KS patients [9,21]. Therefore, patients who suffer impairments in spermatogenesis before completing puberty, like pediatric cancer survivors and KS, urgently need alternative methods to overcome infertility. 

It has been suggested that prepubertal patients at risk of male infertility with no available sperm could undergo early testicular tissue cryopreservation to save spermatogonial stem cells (SSC) that could be used in future experimental fertility treatments [12,22]. This initiative is currently in place in many different centers worldwide, and their leading target groups are prepubertal boys undergoing gonadotoxic chemotherapy, bilateral un-descendent testis, or KS [23,24,25,26,27,28]. Many SSC-based fertility therapies are being explored with promising preliminary results, including in vitro spermatogenesis [29,30], SSC transplantation [31,32], testicular tissue grafting [33,34,35], and testicular ex vivo maturation [36,37,38,39,40,41]. In vitro spermatogenesis has been considered a possible solution for these patients. However, despite many efforts from different research groups, a complete in vitro spermatogenesis system needs to be available for humans [42,43,44]. 

SSC isolation and in vitro propagation from cryopreserved testicular tissue have been well-established for both prepubertal XY and XXY boys [45,46,47,48,49], paving the way for clinical application in the future. In previous work by our group, HTOs were successfully created using cells isolated from mature testicular tissue [29]. In these studies, HTOs maintained viability and morphology over a 3-week period in 3D culture, producing androgen both with and without pulsed hormonal stimulation. Moreover, gene expression analysis and immunostaining for postmeiotic markers showed spermatogonia maturation towards postmeiotic germ cells. HTOs also represent a scalable, high-throughput 3D human testicular in vitro model suitable for toxicity [29] and drug discovery studies [50,51]. In addition, this model was used to study the pathophysiology of testicular diseases such as chronic Zika virus infection or COVID-19 [52,53].

This current study builds on our previous organoid system towards potential clinical application. The primary modification was that all cells, including Sertoli and Leydig cells, were primary cells, and no immortalized cells were used in this study. We isolated and propagated primary testicular cells from immature testicular tissue samples collected from a prepubertal XY and two peripubertal KS patients to form a 3D HTO for in vitro spermatogenesis. 

## 2. Materials and Methods

### 2.1. Human Testis Material

Immature human testicular tissue (whole testis) from a 10-year-old brain-dead patient was acquired through the National Disease Research Interchange (NDRI). KS testicular tissue biopsies were donated by 15 and 17-year-old KS patients enrolled in the Atrium Health Wake Forest Baptist testicular bank under IRB-approved protocols at Wake Forest School of Medicine (IRB00021686 and IRB00061265).

Testicular tissues were formalin-fixed for histological examination. The remaining testicular tissue was processed into small pieces, and slow freezing cryopreservation was performed [23]. Cryotubes were transferred to vapor nitrogen (−196 °C) for long-term storage.

### 2.2. Morphologic Testicular Tissue Evaluation

Preliminary analyses were conducted to evaluate the differentiation stage of testicular tissue prior to isolation and culture. Tubules with spermatogonia were expected from XY prepubertal donors. XXY peripubertal samples would represent the common clinical scenario where no sperm is found in KS patients undergoing micro-TESE. Small pieces of tissues were fixed in 4% paraformaldehyde (PFA) and Bouin’s fixative solution, then paraffin-embedded following an established protocol. Hematoxylin and eosin staining was performed on 5 µm Bouin fixed sections using an autostainer (Leica ST5010 Autostainer XL, Leica Biosystems, Deer Park, IL, USA) to evaluate the testis morphology, including the size of the seminiferous tubules and the presence of spermatogenesis. Microscopic images were acquired using a LEICA DM4000B microscope (Leica Biosystems, Deer Park, IL, USA), Olympus camera DP73 (Olympus America Inc, Center Valley, PA, USA), and Olympus Cellsens software (Version 4.1.1 64bit).

Immunohistochemical staining for protamine-1 (PRM1), a postmeiotic germ cell marker, was performed on PFA fixed tissue. Briefly, 5 µm sections were deparaffinized using an autostainer, and antigen retrieval was accomplished using Na-citrate 0.01 M (pH = 6.0) at 98 °C for 30 min. After washing in 1X PBS 3 times, permeabilization was performed using 0.2% Triton-X-100 for 10 min. Dako serum-free protein block reagent (Dako, Catalogue #X090930-2) was used to block non-specific binding for 30 min at room temperature (RT). The excess protein block was removed, and sections were incubated with primary antibody (rabbit anti-human PRM-1, 1:100 dilution, Sigma #HPA055150) overnight at 4 °C. Normal rabbit anti-human IgG (1:100 dilution, SC-2027, Santa Cruz Biotechnology, Santa Cruz, CA, USA) was used as a negative control, and mature human testicular tissue was used as a positive control. The next day, slides were rinsed in 1X PBS 3 times. Avidin/Biotin blocking kit (#SP-2001 Vector Laboratories, Newark, CA, USA) was used for 45 min. Sections were incubated with goat anti-rabbit biotinylated antibody (1:200 dilution) as a secondary antibody for 1 h at RT. Slides were rinsed in 1X PBS 3 times and then incubated with Streptavidin-Alexa Fluor 594 (1:200 dilution) for 1 h in darkness at RT. After rinsing in 1X PBS 3 times, slides were counterstained using Vectorshield mounting media with 4′, 6-diamidino-2-phenylindol (DAPI) (#H-1200, Vector Laboratories, Newark, CA, USA) and stored at −20 °C. 

Immunohistochemical staining for PGP 9.5 (UCHL1) (PA0286; mouse monoclonal), a non-differentiated spermatogonia marker, and hematoxylin counterstaining was performed following a previously reported in-house protocol [47]. Images were taken using a Leica DM4000B microscope (Leica Biosystems, Deer Park, IL, USA) and QImaging Retiga 2000 RV camera (Teledyne Photometrics, Tucson, AZ, USA). 

### 2.3. Cell Isolation, Propagation, Characterization, and Quantification

#### 2.3.1. Isolation and Propagation 

Previously, cryopreserved 1–2 mg pieces of testicular tissue pieces were slowly thawed in 37 °C running water. Testicular cell isolation, including enrichment for spermatogonia stem cells (SSCs), was performed using a two-step mechanical and enzymatic digestion system previously described [45,46,47,49]. For this study, clinically approved collagenase (NB 4 Standard Grade SERVA Electrophoresis, Heidelberg, Germany) and protease (Natural Protease NB, SERVA Electrophoresis, Heidelberg, Germany) were used to facilitate the translation of our method from research to the clinical setting. 

Isolated cells were cultured on uncoated plastic plates at a seeding density of 10,000–20,000 cells/cm^2^ in supplemented 1X MEM (1X MEM with 10% FBS, 1X non-essential amino acids (Invitrogen, Carlsbad, CA, USA), 15 mM HEPES (Invitrogen, Carlsbad, CA, USA), 50 µg/mL gentamicin (Invitrogen, Carlsbad, CA, USA), 4 mM L-glutamate (Invitrogen, Carlsbad, CA, USA) 0.12% sodium bicarbonate, streptomycin (100 µg/mL), penicillin (100 IU/mL) (Sigma-Aldrich, Burlington, MA, USA)) overnight. The next day, media were changed to supplemented StemPro-34 (Gibco-Thermo Fisher Scientific, Waltham, MA, USA) media (StemPro-34 with recombinant human GDNF (40 ng/mL) (Sigma-Aldrich, Burlington, MA, USA), recombinant human EGF (20 ng/mL), recombinant human Leukemia inhibitory factor (10 ng/mL), streptomycin (100 μg/mL), penicillin (100 IU/mL) (Sigma-Aldrich, Burlington, MA, USA)) (Table 1). 

Media were refreshed every four days, and cells were passaged when 80% confluent (approximately every 7–10 days). Somatic cells (mainly Sertoli, Leydig, and peritubular cells) from the initial tissue sample were utilized as a feeder layer for proliferating germ cells [49]. Excess cells after each passage were cryopreserved as a backup at −196 °C in 1X MEM containing 20% FBS and 8% DMSO.

#### 2.3.2. Characterization of the Cells in Culture

We characterized the peripubertal XXY cells propagated in vitro as previously published [47]. To characterize the XY prepubertal cells in our two-dimensional (2D) culture, 200 K cells were snap-frozen in liquid nitrogen, RNA was isolated using RNeasy Mini Kit (QIAGEN, Venlo, The Netherlands) and later converted to cDNA using a cDNA Reverse Transcription Kit (Life technologies, Carlsbad, CA, USA). For reverse transcriptase quantitative PCR (qRT-PCR) analysis, Taqman^®^ gene expression assays (Thermo Fisher Scientific, Waltham, MA, USA) were used. Germ cell gene expression markers, including undifferentiated spermatogonia: *PLZF (ZBTB16)*, *UCHL1 (PGP 9.5)*, and *THY1 (CD90)*; differentiating spermatogonia: DAZL; and postmeiotic germ cell: *PRM1* and *ACROSIN*, were evaluated. The following somatic cells markers were used for Sertoli cells: *CLUSTERIN*, *SOX9*, and *GATA4*; Leydig cells: *STAR*, *TSPO*, and *CYP11A1*; and for peritubular cells: *CD34* and *ACTA2* (Table 2). 

All reactions were performed using standard Taqman^®^ Universal PCR Master Mix (Thermo Fisher Scientific, Waltham, MA, USA) and run on an ABI-7300 FAST System (Applied Biosystems-Thermo Fisher Scientific, Waltham, MA, USA). For each reaction, 25–50 ng of cDNA was used. Cycling conditions were as follows: 95 °C for 10 min, 95 °C for 15 s (40 cycles), and 60 °C for 1 min. All runs were performed in triplicate, and expression of all genes were normalized to the RNA polymerase II subunit gene (housekeeping gene, POLR2A), as an internal control. Relative expression was determined using the 2^−∆∆CT^ method. We compared the relative expression of these genes with an adult testicular tissue sample as a positive control. 

PCR products were later run on a 3% agarose gel in TAE buffer at 120 mA for 20–30 min. Pictures of the bands were obtained using the Kodak Gel Logic 200 Imaging system (Kodak, Rochester, NY, USA). 

#### 2.3.3. SSC Identification 

Although a specific marker for SSCs has not yet been identified, the combination of HLA-ABC negative with CD9+, CD90+, FGF-R3+, and SSEA4+ surface markers has been proven to enrich for SSC [54,55,56,57,58,59]. Flow cytometry was used to assess the putative SSC population in culture. Mouse IgG for each condition was used as an isotype control (Table 3). 

Following trypsinization and counting, 100 K for each marker/condition was washed with flow cytometry buffer (PBS + 0.5% FBS). Cells were pelleted, resuspended in the buffer, and aliquoted into flow cytometry tubes with 100 K in 100 µL of buffer in each tube. An amount of 10 µL of the respective antibody was used for each condition and kept in the dark for 20 min at RT. Cells were then washed, pelleted, and resuspended in buffer (100 µL per condition) and then kept on ice until flow cytometry was performed using the BD Accuri C6 machine (BD Accuri, Franklin lakes, NJ, USA). Ten thousand events were recorded for each group. Gating, compensation, and analysis were performed using BD Accuri C6 software (version 227.4) and FCS Express 7 (version 7.22) 

### 2.4. Human Testicular Organoid (HTO) Formation and Differentiation

Cells in culture were passaged and seeded in organoid formation media (Table 1) supplemented with human testis extra cellular matrix (ECM) [60], into 96-well ultra-low attachment round-bottom plates (Corning Costar Ultra-low Attachment Multiwell U-bottom #7007, Corning, NY, USA) using 10,000 cells per organoid. Plates were centrifuged at 150× *g* for 30 s. Cells were cultured at 37 °C under 5% CO_2_ for 48 h until organoid formation was completed. HTOs were then refreshed with differentiating media (Table 1) every other day and kept for three weeks in culture at 34 °C under 5% CO_2_. HTO size, viability, ATP production, gene expression, number of cells, histological morphology, testosterone production, and chromosomal ploidy were analyzed at four different time points: 2, 9, 16, and 23 days in culture. 

#### 2.4.1. HTO Diameter and Viability

Molecular probes live–dead cell imaging (Invitrogen, Carlsbad, CA, USA, L3224) was used to evaluate HTO viability. Calcein AM (green, live cells) and ethidium homodimer (red, dead cells) were added to HTO media and incubated at 37 °C for 20 min. After the excess dye was washed out with 1X PBS, HTOs were imaged and diameters measured using a Leica TCS-LSI Macro Confocal microscope (Leica Biosystems, Deer Park, IL, USA).

#### 2.4.2. ATP Production

At every time point, 8 HTOs were harvested to assess ATP production using an ATP kit (CellTiter-Glo Luminescent Cell Viability Assays, Promega, Madison, WI, USA) following the manufacturer’s instructions. The luminescence signal was measured in relative light units (RLUs) with a Turner Biosystems Veritas Microplate Luminometer (#9100-002 Promega, Madison, WI, USA).

#### 2.4.3. Cell Dissociation and RNA Isolation 

At each time point, 96 HTOs were enzymatically digested in collagenase (NB 4 Standard Grade SERVA Electrophoresis, Heidelberg, Germany) and protease (Natural Protease NB, SERVA Electrophoresis, Heidelberg, Germany) for 2 h on a shaking rotator with 120 RPM at 37 °C (similar to cell isolation). Additional mechanical dissociation was then performed by repeated pipetting until single cells were obtained. The dissociated cells were counted using a hemocytometer, and the ratio of cells retrieved per HTO was calculated. Cells were then resuspended in 350 µL of RLT/BME and snap-frozen in liquid nitrogen for RNA extraction using an RNeasy QIAGEN Mini Kit (Venlo, The Netherlands).

#### 2.4.4. Histology Evaluation 

At every time point of the study, 24 HTOs were pooled and fixed in 4% PFA for 30 min at room temperature. Fixed HTOs were embedded in HistoGel (Thermofisher, Waltham, MA, USA HG-4000-012) and paraffinized following an in-house protocol. Hematoxylin and eosin staining (HE) were performed on 5 μm sections using an autostainer. Pictures of the HTOs were captured on a Leica DM400 B LED microscope (Leica Biosystems, Deer Park, IL, USA).

#### 2.4.5. Testosterone Production 

HTO ability to produce androgen was evaluated by the Wake Forest Baptist Health Medical Center clinical laboratory. A total of 32 HTOs were analyzed per time point, half of these were stimulated with ten mIU of hCG for 3 h. HTOs were then harvested, and media were pooled and stored at −80 °C. Testosterone in the HTO media was measured by competitive binding immunoenzymatic assay (Beckman DXI800 analyzer, Beckman Coulter, Brea, CA, USA). The testosterone level was compared between pulse-stimulated and non-stimulated HTOs at each time point. Fresh differentiation culture media were used as a negative baseline control.

#### 2.4.6. Gene Expression Analysis

Isolated RNA from HTOs was converted to cDNA using a high-capacity Reverse Transcription Kit (Life Technologies, Carlsbad, CA, USA). Quantitative reverse transcriptase PCR (qRT-PCR) analysis and Taqman^®^ gene expression assays were used to evaluate testicular cell type-specific gene expression changes over time (Table 2). Reactions were performed using standard Taqman^®^ Universal PCR Master Mix (Thermofisher, Waltham, MA, USA (96-well plate format) and run on an ABI 7300 FAST system (Applied Biosystems-Thermo Fisher Scientific, Waltham, MA, USA). Cycling conditions were as follows: 95 °C for 10 min, 95 °C for 15 s (40 cycles), and 60 °C for 1 min. During data analysis, the expression of all genes was normalized to the POLR2A housekeeping gene; relative expression was calculated using the 2^−ΔΔCT^ method. All runs were performed in triplicate.

#### 2.4.7. DNA Fluorescence In Situ Hybridization (FISH) of the HTOs for Ploidy Identification 

Sections of 5 µm were prepared for histological evaluation. FISH for 18, X, and Y chromosomes was performed on consecutive sections using an AneuVysion Multicolor DNA Probe Kit (Vysis CEP 18/X/Y—alpha satellite from Abbot, Abbot Park, IL, USA). Nuclei were counterstained with DAPI. Images of the hybridized slides were taken using a fluorescence filter microscope (Zeiss Axiophot microscope Dublin, CA, USA) and image acquisition software (Applied Spectral Imaging Software version 8.3.2). Due to the color similarities of X (green) and 18 (blue) probes, the color of chromosome 18 probe was digitally converted to yellow (pseudo color). FISH signals were analyzed in consecutive sections among different HTOs. The presence of only one X or one Y accompanied by one 18 chromosome in a nucleus along all consecutive sections was considered a haploid cell.

### 2.5. Statistics

Statistical analysis of all quantitative results is presented as mean ± standard deviation (SD). Statistical significance was determined via GraphPad Prism version 8.0.2 software using a Student *t*-test, with *p* values < 0.05 considered statistically significant.

## 3. Results

### 3.1. Immature Testicular Tissue with Neither Differentiating nor Differentiated Germ Cells 

Morphologic analysis using HE staining on prepubertal XY donor testes showed small seminiferous tubules with no signs of active spermatogenesis. Peripubertal KS patients showed severe alteration of testicular architecture with signs of established testicular fibrosis and few seminiferous tubules present with no apparent active spermatogenesis. Conversely, adult XY control testicular tissue demonstrated prominent seminiferous tubules with active complete spermatogenesis (Figure 1).

Immunohistochemical staining was performed to further characterize the testicular samples used in this study. The postmeiotic marker protamine-1 (PRM1) was used in the prepubertal samples to identify loci of active spermatogenesis. On the KS samples, the undifferentiated spermatogonia marker PGP 9.5 (UCHL1) was used to confirm the presence of seminiferous tubules with preserved spermatogonia. No PRM1-positive cells were found in the prepubertal donor samples (Figure 2) while they were present in great numbers within the mature control testicular tissue, confirming the immature stage of the prepubertal testicular samples. PGP 9.5 was prominently expressed in the XY age-matched control, while only a few positive cells were identified in KS testicular tissue samples. (Figure 2).

### 3.2. Long-Term Propagation of Testicular Cells

To isolate cells, previously cryopreserved tissue was mechanically disassociated and enzymatically digested according to established methods for SSC culture with some modifications, as described in the methods section above. Cells were cultured in a 2D system supplemented with StemPro-34 media (Table 1). Cells isolated from 10-year-old XY prepubertal donor right and left testicles were propagated in culture for 5 passages (53 days), and 3 passages (36 days), respectively. Cells isolated from 15- and 17-year-old KS peripubertal patients were propagated for 4 passages (34 days), and 3 passage (31 days), respectively. Media were refreshed every 2–3 days, and cells were passaged at 80% confluency.

### 3.3. Presence of All Major Cell Types in Propagated Prepubertal Testicular Cells In Vitro

To reproduce normal testicular physiology, in vitro, the main testicular cell types should be present in culture. qRT-PCR analysis was used to identify characteristic gene expression markers ZBTB16, UCHL1, and THY1 for undifferentiated spermatogonia; GATA4, SOX9, and Clusterin (CLU) for Sertoli cells; STAR, TSPO, and CYP11A1 for Leydig cells; and CD34 and ACTA 2 for peritubular cells. Positive gene expression confirmed the presence of all four cell types in the 2D prepubertal testicular cell culture (Figure 3). These findings are consistent with the previously reported characterization of cultured XXY testicular cells [47].

The presence of a putative SSC population in culture was assessed using flow cytometry for frequently used cell surface markers. Depending on the specific marker combination, prepubertal XY cultured cells in passage 3 showed an SSC population of between 2.3% and 11.4% of the total population (Figure 4). Similar findings were previously reported on KS testicular cells in 2D culture [47]. These data confirmed the presence of the SSC population in our 2D culture system confirming the potential to colonize seminiferous tubules and initiate spermatogenesis. 

### 3.4. HTO Formation and Germ Cell Differentiation

HTOs were formed in ultra-low attachment round-bottom plates using the protocol detailed in the methods section (Figure 5). A video of the formation process using the IncuCyte Zoom live-cell analysis system (Essenbioscience-Sartorious, Göttingen, Germany) is included in the Supplemental Data (Appendix A). 

#### 3.4.1. HTOs Maintain Their Structure, Viability, and Function for Three Weeks in Culture 

HTO morphology demonstrated cohesively packed cells with prominent nuclei (Figure 6). Live/dead staining indicated robust viability throughout the culture (Figure 6). Core necrosis was occasionally observed at the last point of the study, 21 days, in some technical replicants (Figure 6). 

The size and ATP production (Figure 7) in HTOs were monitored weekly and showed a statistically significant drop after the first week of culture. Viability stabilized after the first week and was maintained throughout the remainder of the study. No significant differences in size or ATP production were observed between XY and XXY HTOs. 

#### 3.4.2. Testosterone Production by HTOs

At each weekly time point, testosterone production was detected in both hCG-stimulated and not-stimulated HTOs, showing that they could consistently produce testosterone, in vitro (Figure 8). No significant difference was observed in basal testosterone production between XY and XXY HTOs (Figure 8). hCG stimulation led to a significant increase in testosterone production at the first time point in prepubertal XY cells. These results were not observed at later points. Conversely, hCG stimulation did not significantly increase testosterone production in KS HTOs at any time point (Figure 8).

#### 3.4.3. Germ Cells Differentiation of HTOs

Gene expression analysis showed a significant upregulation of non-differentiated (DAZL) and postmeiotic germ cell markers (Acrosin and PRM1) between day 2 and day 23 in culture (Figure 9a). When gene expression analysis was evaluated using a heat map plot at each time point, a transitory upregulation of intermediate meiosis marker SYCP 3 was identified (Figure 9b). Moreover, digital PCR indicated a population of at least 0.2% expressing PRM1 and a population of at least 0.2% cells expressing acrosin at the latest time point, indicating the presence of spermatid cells (Figure 9c). No significant differences were observed between prepubertal and Klinefelter HTO gene expression levels.

#### 3.4.4. Presence of Haploid Cells inside Differentiated HTOs

Finally, fluorescence in situ hybridization (FISH) for X, Y, and 18 chromosomes was performed on fixed HTOs. While the surface cells were diploid, haploid cells were identified in the core of both prepubertal and Klinefelter HTOs after 23 days in culture (Figure 10A,C). No haploid cells were observed at earlier time points. Quantification of the FISH signal showed 6.42% of 18,X haploid cells and 6.68% of 18,Y haploid cells inside the prepubertal XY HTO. In contrast, prepubertal Klinefelter HTOs showed 10.9% of 18X haploid cells and 4.6% of 18Y haploid cells (Figure 10B,D). 

## 4. Discussion

This study’s first goal was to prove the feasibility of forming viable and stable testicular organoids from prepubertal 46,XY and Klinefelter testicular cells. Live/dead staining showed good viability at every weekly time point, and only mild core necrosis was observed at the end of the three-week study (Figure 6). On the other hand, considering ATP production analysis as an indirect method to assess viability, a significant drop was found after the first week, which stabilized over the remainder of the study. 

The transition from a 2D propagation culture into a 3D differentiation system may modify the physiologic energy requirements and impact ATP production without compromising cell viability. Another plausible explanation is that although cells in the HTOs remained viable, the overall number of cells decreased over time. In preliminary studies comparing different organoid seeding concentrations, the results showed a consistent 33% incorporation rate of cells forming HTOs, regardless of seeding concentration, while maintaining excellent viability. The washout of cells not incorporated into the HTO structure would account for the initial ATP drop. 

Some cells incorporated into the HTO structure may die during the initial structural remodeling process. The remnant cellular cytoskeleton and matrix from cells lost during this process may have improved HTO cohesion and helped shape the 3D structure. This observation is supported by the amount of “conjunctive tissue-like” material seen in the HE histology images. Viability assay results were comparable, with no significant difference between prepubertal 46,XY and KS HTOs. This suggests that 3D in vitro culture of KS testicular cells may bypass and overcome the germ cell depletion and testicular fibrosis observed in vivo. 

The testes play a critical role in human physiology through both androgen secretion and male gamete production. The ultimate goal in producing HTOs is to reproduce testicular physiology, in vitro*,* for clinical and research applications. Therefore, a cornerstone of our study was to assess testosterone production in the 3D HTOs with and without hCG pulse stimulation. Both prepubertal 46,XY and KS HTOs secreted significant levels of testosterone at every time point of the study (Figure 8). It is important to mention that no testosterone was added to the culture media, but a basal concentration of hCG was added (Table 1). Moreover, immediately following formation, prepubertal HTOs stimulated by hCG secreted a significantly higher amount of testosterone than non-stimulated controls, mirroring results expected in vivo and consistent with previously reported findings [29].

However, with longer time in culture, the differences between stimulated and non-stimulated HTOs were not significant. A possible explanation for these results is that Leydig cells became desensitized to hCG stimulation under differentiation conditions. Another reason could be that the decrease in overall cell number resulted in too small a population of Leydig cells to measure significant differences. Nevertheless, the most obvious explanation would be that the stimulation method used in this study is not sufficient to induce testosterone production in prepubertal testes. Future studies will explore and compare different stimulation patterns and culture media to optimize testosterone production in the HTO system.

Despite producing testosterone in culture, KS HTOs did not significantly respond to hCG pulse stimulation (Figure 8). This may indicate that KS HTOs mimic the disease phenotype, and KS Leydig cells do not respond to hCG to a degree comparable to 46,XY prepubertal and adult controls. 

Following previous work from our group on adult cell HTOs [29], to our knowledge, this is the first study reporting in vitro testosterone production from both prepubertal 46,XY and Klinefelter testicular cells. Reports from Sun et al. [30] presented a testicular organoid model entirely focused on spermatogenesis with no Leydig cell incorporation. However, our group envisions HTOs as being similar in composition to human testes, in vivo. Therefore, Leydig cells and testosterone production are critical. Including steroidogenic cells in our 3D culture system could add value to spermatogenesis, in vitro*,* in more ways than anticipated. Moreover, the successful recreation of steroidogenic pathways, in vitro*,* could open the door to other possible HTO clinical applications, including hormone replacement therapy. 

Gene expression analyses were conducted to identify signs of in vitro spermatogenesis in the cells in culture. The results showed the transition from undifferentiated spermatogonia to postmeiotic spermatids over the three-week differentiation period, mimicking in vivo findings (Figure 9). Initially, cells in the 2D culture system and testicular tissue showed no differentiated spermatogonia expression, and the undifferentiated spermatogonia marker (ZBTB16) was highly expressed (Figure 3). When cells were included in the 3D culture system, undifferentiated spermatogonia (ZBTB16) and differentiating spermatogonia (DAZL) markers steadily increased expression. HTOs began expressing meiosis markers (SCYP3) after one week, and expression levels peaked at week two and dropped at week three (Figure 9). Finally, after three weeks in culture, positive expressions of postmeiotic markers (PRM1 and Acrosin) were detected (Figure 9). These findings are consistent with the known spermatogenesis gene expression pattern. Interestingly, an increase in the undifferentiated spermatogonia gene expression marker (ZBTB16) was also observed. One explanation may be that not all the spermatogonia cells in our 3D system are synchronized. It is possible that some spermatogonia cells differentiate while others prepare for the next spermatogenesis wave. This would also explain why the meiosis marker SCYP3 peaked by week two and then declined.

Using digital PCR, the postmeiotic cell population was estimated to represent at least 0.2% of all HTO cells. These data will allow us to estimate the number of HTOs needed to produce enough haploid cells for clinical applications. After three weeks, haploid cells with either X or Y chromosomes were identified through FISH staining inside HTOs. No haploid cells were observed at earlier time points of the study. Previous work by our group provided evidence of in vitro maturation of spermatogonia to the postmeiotic stage using a 3D culture system [29] comprised of adult testicular cells. Although this represented suitable proof of concept for our current work, some questions arose regarding the use of adult testicular tissue. We questioned whether cells included in the 3D system from adult human testes were already committed to spermatogenesis and the HTOs simply supported this process. Alternatively, the 3D system may push and accelerate spermatogonia into differentiation and shorten the spermatogenesis cycle in vitro. Similarly, a recent report by Sun et al. [30] used cells from a patient diagnosed with obstructive azoospermia for in vitro spermatogenesis, and complete spermatogenesis was confirmed. In this study, the possibility of having differentiated spermatogonia is negated by using peripubertal testicular tissue without active spermatogenesis, as confirmed both by clinical embryology and pathology laboratories. Hence, it can be safely concluded that any differentiation observed in this study occurred strictly in vitro and could potentially represent a future clinical option for non-obstructive azoospermic patients. 

This study sought to bring the HTO system closer to clinical application. Consequently, immortalized Sertoli and Leydig cells used in our previous work [29] were replaced by primary somatic cells. Only GMP (good manufacturing practices)-qualified digestive enzymes (Collagenase NB 4 Standard Grade and Natural Protease NB SERVA Electrophoresis, Heidelberg, Germany) were used for cell isolation. The KS testicular tissue came from an experimental testicular tissue bank for fertility preservation. As only a small portion of the stored material (20%) was used for research, the same patients could benefit from the techniques developed in this study using the remaining tissue (80%) to restore fertility. 

Previous studies in mice have used an organotypic 3D culture system for successful in vitro maturation of spermatogonia, leading to fertile offspring [61,62]. In another study using human testicular cells from obstructive azoospermic human patients, haploid cells capable of fertilizing mouse eggs were produced in vitro [30]. However, these studies identify knock out serum replacement (KSR) as a critical factor in cell differentiation. In our studies, we avoid using KSR due to concerns about the known impact on chromosome stability [63]. In the same way, Sun et al. [30] meticulously selected spermatogonia and Sertoli cells with a high level of purity to improve spermatogenesis efficiency. However, this study included Sertoli, Leydig, and peritubular cells, as they may help support spermatogenesis and better recreate the overall human testicular physiology. 

Although Klinefelter syndrome is considered the most common genetic cause of male infertility, its pathophysiological mechanism remains widely unknown. The fact that our 3D organoid culture system replicates both meiosis and androgen production, in vitro, supports its suitability as a Klinefelter syndrome disease model. Therefore, the HTO model could play a key role in better understanding Klinefelter syndrome and how to address it clinically.

This study presents promising evidence of in vitro germ cell differentiation. With recent advances in assisted reproductive techniques such as round spermatid injection (ROSI) or elongated spermatid injection (ELSI), future studies exploring the fertilization capability of in vitro*-*produced haploid cells may unveil this system’s potential. This method would be especially valuable in cancer survivors where the potential risk of re-introducing malignant cells to the patient may preclude the use of SSC transplantation or testicular tissue grafting. By acquiring the haploid gamete through in vitro maturation and consequently utilizing it for ROSI, ELSI, or ICSI, any malignancy threat would be avoided. This work will also push in vitro spermatogenesis a step closer to clinical application for KS patients where no sperm are retrieved via microTESE.

## 5. Conclusions

To the best of our knowledge, this is the first report of in vitro haploid cell differentiation and androgen production using human testicular organoids from prepubertal and Klinefelter patients’ cells.

## Figures and Tables

**Figure 1 bioengineering-11-00677-f001:**
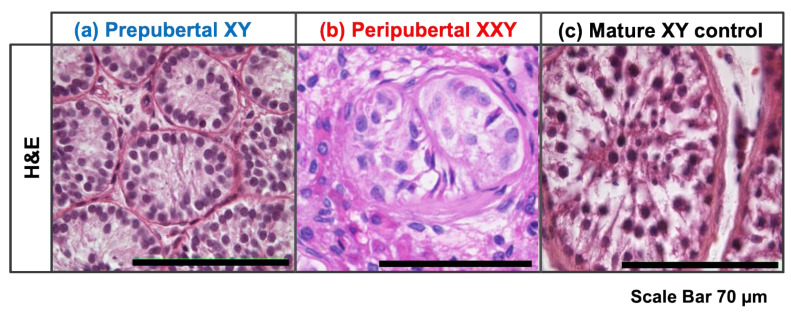
Hematoxylin and eosin (HE) staining on histology testicular tissue slides from (**a**) prepubertal XY donor, (**b**) Klinefelter syndrome peripubertal patient, and (**c**) XY mature control.

**Figure 2 bioengineering-11-00677-f002:**
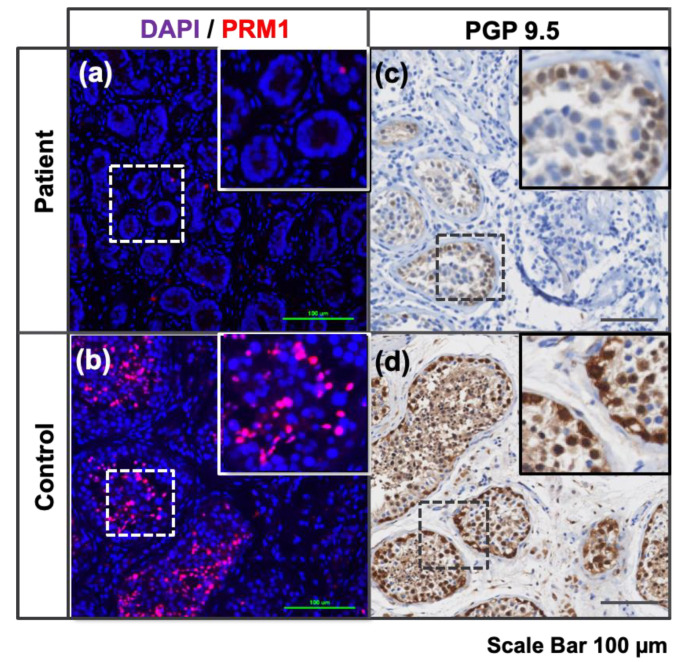
Immunostaining of testicular tissue. (**a**,**b**) Immunofluorescence for the protamine-1 (PRM1) early postmeiotic marker shows late-stage round spermatid (red) and nucleus (DAPI, blue) on (**a**) prepubertal donor tissue samples and (**b**) adult testis control. The differences in seminiferous tubules structure and the absence of PRM-1 positive cells in prepubertal patient sample confirmed the immaturity of the tissue. (**c**,**d**) Optical immunostaining for the undifferentiated spermatogonia marker PGP 9.5 (UCHL1) was demonstrated by the DAB chromogen (brown) and hematoxylin counterstain (blue) on (**c**) Klinefelter peripubertal tissue samples and (**d**) age-matched XY.

**Figure 3 bioengineering-11-00677-f003:**
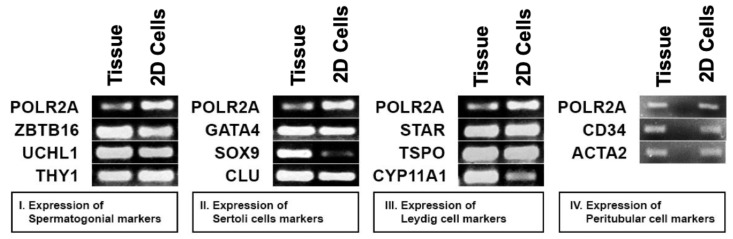
Characterization of 2D cultured prepubertal testicular cells with qRT-PCR for different cell-type markers. (**I**) Spermatogonial cell markers: ZTBT16, UCHL1, and THY1. (**II**) Sertoli cell markers: GATA4, CLU (Clusterin), and SOX9. (**III**) Leydig cell markers: STAR, TSPO, and CYP11A1. (**IV**) Peritubular cell markers: CD34 and ACTA2 on whole testis and isolated cells from the study patient. POLR2A (DNA-directed RNA polymerase II subunit RPB1) was utilized as a reference marker in all cell types. All the primers were tested to be exon spanning, both genomic DNA and water were used as negative controls with no band shown (no amplification).

**Figure 4 bioengineering-11-00677-f004:**
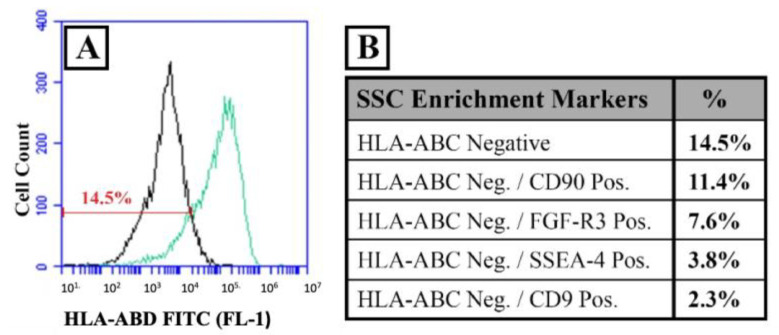
Flow cytometry using enriching markers for spermatogonia stem cells (SSC) in 2D cultures of human prepubertal testicular cells. (**A**) Histogram of HLA-ABC FITC-conjugated positive cells (green) compared to isotype control (black). (**B**) Table of percentages of positive cells for different SSC enrichment markers combined: CD90, FGF-R3, SSEA-4, and CD9 without HLA-ABC.

**Figure 5 bioengineering-11-00677-f005:**
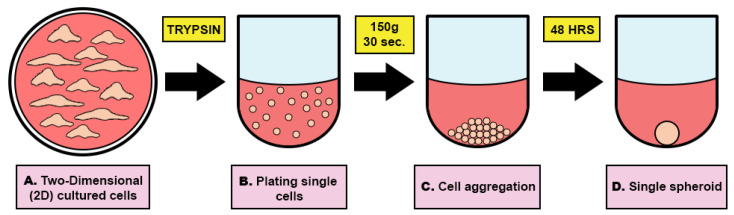
Schematic picture illustrating the creation of three-dimensional (3D) testicular organoids from cultured testicular cells. (**A**) Two-dimensional cultured cells were trypsinized and made single cells in a suspension of formation media. (**B**) Single cells were plated in 10,000 cells/100 µL/well. (**C**) Plates were centrifuged at 150× *g* for 30 s to make cell aggregates. (**D**) After 48 h, single spheroids were formed.

**Figure 6 bioengineering-11-00677-f006:**
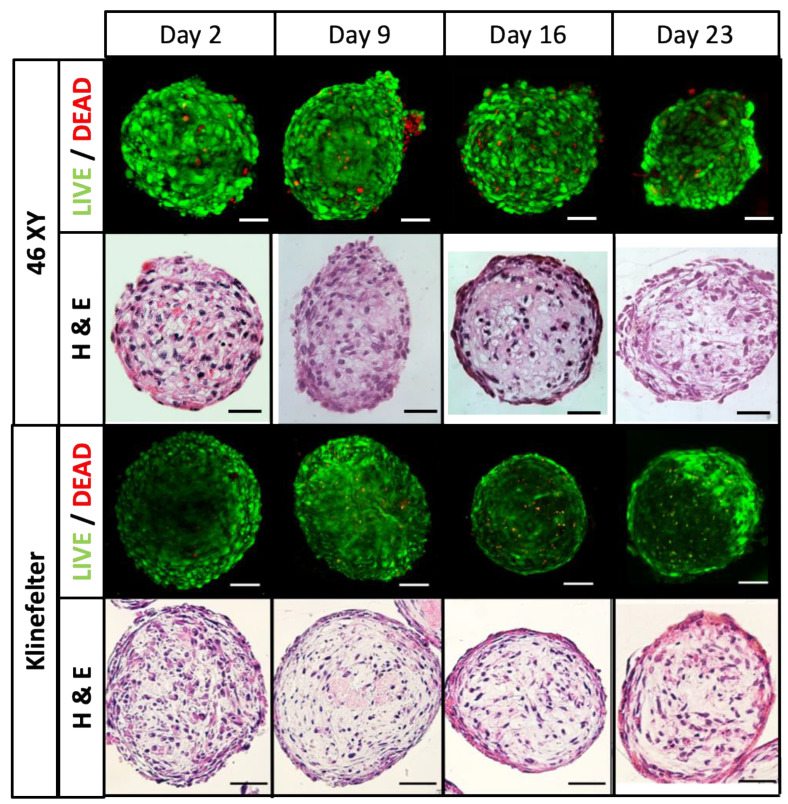
Prepubertal (top two rows) and Klinefelter (bottom two rows) HTO morphology and viability. For each condition, the upper row represents live/dead staining, and the lower row is HE staining at each study time point. Scale bars 100 μm.

**Figure 7 bioengineering-11-00677-f007:**
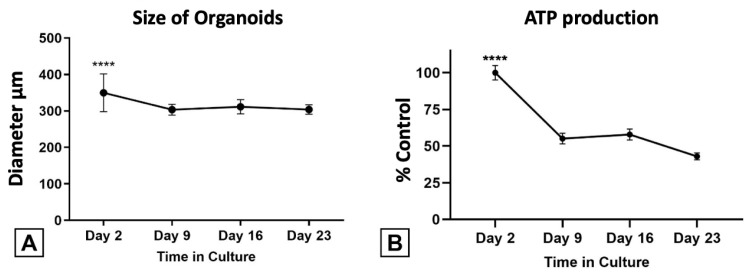
Evolution of the HTOs in culture. As no difference was observed between 46XY and 47XXY organoid size and ATP, results show data averages. (**A**) Average size at every time point of the study. (**B**) Average ATP production compared to initial determination. Data presented as mean ± SD. Significance: **** *p* < 0.000. Data showed the overall average of 46XY and KS HTOs as the two groups had no significant differences. Negative control: HTO medium without ATP assay regents was used to subtract the background signal.

**Figure 8 bioengineering-11-00677-f008:**
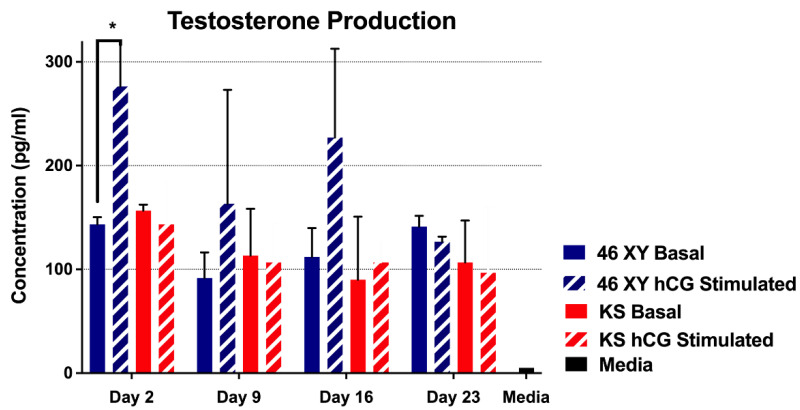
Testosterone production of the HTOs in culture with or without previous hCG pulse stimulation. Data presented as mean ± SD. Significance: * *p* < 0.05.

**Figure 9 bioengineering-11-00677-f009:**
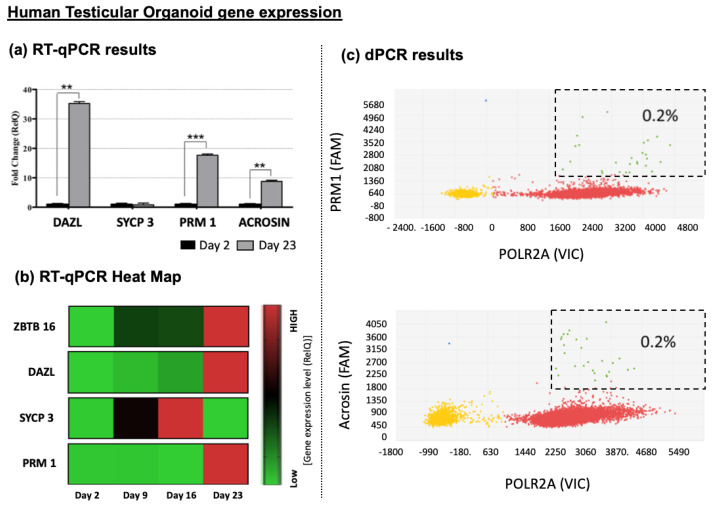
Gene expression analysis of the HTO in differentiation culture media over time using qPCR-RT and dPCR. The data show no significant differences were found. (**a**) Gene expression comparison between HTOs on day 2 and day 23 in culture expressed as a fold increase in expression level. (**b**) Heat map of gene expression for germ cells at sequential differentiation stages: undifferentiated spermatogonia (ZBTB16), differentiating spermatogonia (DAZL), differentiating spermatocyte (SYCP3), and differentiated spermatid (PRM1). (**c**) Digital PCR analysis for postmeiotic markers PRM1 and acrosin combined with housekeeping gene POLR2A to assess the proportion of cells expressing these genes by the end of the culture period. Data presented as mean ± SD. Significance: ** *p* < 0.01; *** *p* < 0.001.

**Figure 10 bioengineering-11-00677-f010:**
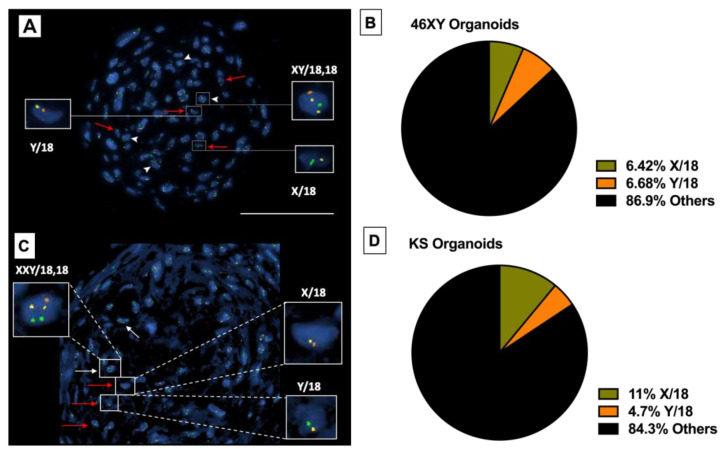
Fluorescence in situ hybridization (FISH) for Cr X (green), Cr Y (orange), and Cr 18 (yellow) was performed on fixed HTOs at every time point. Haploid cells with only one chromosome 18 and one sex chromosome, either X or Y, are identified with red arrows. Diploid cells are identified with white arrows for comparison. (**A**) FISH staining of prepubertal XY HTOs after 23 days in culture. (**B**) Pie chart showing the percentage of the haploid cells amongst the total population of the cells in the prepubertal XY HTOs. (**C**) FISH staining of peripubertal Klinefelter HTOs after 23 days in culture. (**D**) Pie chart showing the percentage of the haploid cells amongst the total population of the cells in the peripubertal Klinefelter HTOs. Scale bar 100 μm.

**Table 1 bioengineering-11-00677-t001:** Three different culture media formulations: testicular cell culture (StemPro complete), testicular organoid formation (formation media), and testicular organoid differentiation media (differentiating media) composition.

Component	Manufacturer	Catalogue #	Final Conc.	StemPro Complete	Formation Media	Differentiating Media
Bovine Serum Albumin	Sigma-Aldrich	A3059	5 mg/mL	✓	✓	✓
D (+) Glucose	Sigma-Aldrich	G7021	6 mg/mL	✓	✓	✓
Ascorbic acid	Sigma-Aldrich	A4544	1 × 10^−4^ M	✓	✓	✓
Transferrin (Apo)	Sigma-Aldrich	T1147	100 µg/mL	✓	✓	✓
Pyruvic Acid	Sigma-Aldrich	P2256	30 mg/mL	✓	✓	✓
d-Biotin	Sigma-Aldrich	B4501	10 µg/mL	✓	✓	✓
2β-mercaptoethanol	Sigma-Aldrich	M3148	5 × 10^−5^ M	✓	✓	✓
DL-Lactic Acid	Sigma-Aldrich	L4263	1 µL/mL	✓	✓	✓
MEM NEAA	Invitrogen	11140-050	10 µL/mL	✓	✓	✓
Stem Pro-34 Supplement	Gibco, Thermofisher	10641-025	26 µL/mL	✓	✓	✓
Insulin	Cell Applications	128-100	25 µg/mL	✓	✓	✓
Sodium Selenite	Sigma-Aldrich	S1382	30 nM	✓	✓	✓
Putrescine	Sigma-Aldrich	P7505	60 µM	✓	✓	✓
L-Glutamin	Invitrogen	25030-081	2 mM	✓	✓	✓
MEM Vitamin Solution	Invitrogen	11120-052	10 µL/mL	✓	✓	✓
Β-Estradiol	Sigma-Aldrich	E2758	30 ng/mL	✓	✓	✓
Progesterone	Sigma-Aldrich	P8783	60 ng/mL	✓	✓	✓
Epidermal Growth Factor (EGF)	Sigma-Aldrich	E9644	20 ng/mL	✓	✓	✓
Human basic Fibroblast Growth Factor (hbFGF)	Sigma-Aldrich	F0291	10 ng/mL	✓	✓	✓
Glial Cell Line-Derived Neurotrophic Factor (GDNF)	Sigma-Aldrich	G1777	10 ng/mL	✓	✓	✓
Leukemia Inhibitor Factor	Sigma-Aldrich	LIF1010	10 ng/mL	✓	✓	✕
Fetal Calf Serum (FCS)	Invitrogen	10437010	1%	✓	30%	✓
Penicilline/Streptomycine (Pen/Strep)	Invitrogen	15140-122	0.50%	✓	✓	✓
Gentamycin	Gibco, Thermofisher	15750078	50 µg/mL	✕	✓	✓
Recombinant human stem cell factor (SCF)	Peprotech	300-07	100 ng/mL	✕	✕	✓
Retinoic Acid	Sigma-Aldrich	R2625	10 µM	✕	✕	✓
Human chorionic gonadotropin (hCG)	Sigma-Aldrich	C8554	1 mIU/mL	✕	✕	✓
Follicle Stimulating Hormone (FSH)	Sigma-Aldrich	F4021	2.5 × 10^−5^ IU/mL	✕	✕	✓
Human testis Extra-Cellular Matrix (ECM)	n/a	n/a	1 µg/mL	✕	✓	✓

Manufacturer location: Sigma-Aldrich (Burlington, MA, USA); Invitrogen (Carlsbad, CA, USA); Gibco, Thermofisher (Waltham, MA, USA); Cell applications (San Diego, CA, USA); Peprotech (Cranbury, NJ, USA).

**Table 2 bioengineering-11-00677-t002:** Taqman^®^ primers used for gene expression analysis.

Gene ID	Amplicon Length (Base Pair)	Catalog #
ACROSIN	144	4331182 (Hs00356147_m1)
PRM1	99	4331182 (Hs00358158_g1)
ZBTB 16 (PLZF)	65	4331182 (Hs00957433_m1)
UCHL 1 (PGP9.5)	80	4331182 (Hs00985157_m1)
THY1 (CD-90)	60	4331182 (Hs009174816_m1)
GATA 4	68	4331182 (Hs00171403_m1)
CLUSTERIN	93	4331182 (Hs00971656_m1)
SOX 9	101	4331182 (Hs01001343_g1)
STAR	85	4331182 (Hs00264912_m1)
TSPO	57	4331182 (Hs00559362_m1)
CYP11A1	81	4331182 (Hs00897320_m1)
CD34	63	4448892 (Hs02576480_m1)
ACTA2	64	4331182 (Hs00909449_m1)
POLR2A	61	4331182 (Hs00172187_m1)4448484 (Hs00172187_m1)

**Table 3 bioengineering-11-00677-t003:** Antibodies used for flow cytometry.

Specificity	Host	Type	Fluorochrome	Manufacturer	Catalog #
HLA-ABC	Mouse	Monoclonal Anti-Human	APC	BD Pharmigen	555555
HLA-ABC	Mouse	Monoclonal Anti-Human	FITC	BD Pharmigen	555552
SSEA-4	Mouse	Monoclonal Anti-Human/Mouse	FITC	R&D Systems	FAB1435F
FGF-R3	Mouse	Monoclonal Anti-Human	PE	R&D Systems	FAB766P
CD-9	Mouse	Monoclonal Anti-Human	PerCP-Cy5.5	BD Pharmigen	341649
CD-90	Mouse	Monoclonal Anti-Human	PE	BD Pharmigen	555596
Control	Mouse	IgG1 Isotype	FITC	BD Pharmigen	340755
Control	Mouse	IgG1 Isotype	PerCP-Cy5.5	BD Pharmigen	347212
Control	Mouse	IgG1 Isotype	APC	BD Pharmigen	340754
Control	Mouse	IgG1 Isotype	PE	BD Pharmigen	340761

Manufacturer location: BD Pharmigen (Franklin lakes, NJ, USA); R&D systems (Minneapolis, MN, USA).

## Data Availability

Data are contained within the article and Appendix A.

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
