# Peer review of "In Vitro Generation of Haploid Germ Cells from Human XY and XXY Immature Testes in a 3D Organoid System"

_bioengineering, 2024, doi:10.3390/bioengineering11070677_

Round 1

Reviewer 1 Report

Comments and Suggestions for Authors

This study developed a 3D human testicular organoid (HTO) system using testicular cells from prepubertal XY and XXY patients to address infertility issues. They showed that the developed HTOs maintained viability, supported testosterone production, and enabled germ cell differentiation, including the formation of haploid cells, indicating potential for SSC-based fertility therapies. It is an interesting study with high potential for clinical application and could generate significant interest in the field. I recommend its publication in Bioengineering after minor revisions to enhance their findings.

1, In Figure 2, the immunostaining of testicular tissue shows a markedly different distribution pattern of the nucleus (DAPI signal) between the control and patient samples. Does the author have any explanations for this difference?

2, In Figure 6, the author did not explain the observations that the 46XY HMOs show increased death in the LIVE/DEAD staining compared to that of Klinefelter group at the same stage. Can the author provide an explanation for this observation?

3, In Figure 6, the representative H&E staining image of the 46XY HTO on day 9 appears different from the others. If this data was collected using different methods, the author should mention this in the methods section and the figure legend.

4, In Figure 7, the author did not specify which group of data was used in panels A and B. For comparison, a negative control is also missing.

5, In Figure 9a, the group labels are missing. Please add the appropriate labels to clarify the data.

6, In Figure 10, the resolution of the images is too low. It would be better to use higher magnification and resolution images to clarify the results.

Author Response

Reviewer 1:

This study developed a 3D human testicular organoid (HTO) system using testicular cells from prepubertal XY and XXY patients to address infertility issues. They showed that the developed HTOs maintained viability, supported testosterone production, and enabled germ cell differentiation, including the formation of haploid cells, indicating potential for SSC-based fertility therapies. It is an interesting study with high potential for clinical application and could generate significant interest in the field. I recommend its publication in Bioengineering after minor revisions to enhance their findings.

1, In Figure 2, the immunostaining of testicular tissue shows a markedly different distribution pattern of the nucleus (DAPI signal) between the control and patient samples. Does the author have any explanations for this difference?

Answer:

Figure 2 (a and b) show Immunofluorescence for the Protamin-1 (PRM1) as an early post-meiotic marker showing late-stage round spermatid (Red) and nucleus (DAPI, blue). (a) and (b) panels represent prepubertal donor tissue samples and adult testis control respectively. The seminiferous tubules of prepubertal males (in panel a) are smaller, narrower, and with few cells per tubule mostly located in the basal membrane or Adluminal. On the other hand, the seminiferous tubules of postpubertal adult controls (b panel) present much bigger, wider tubules filled with germ cells at multiple differentiation stages up to elongated spermatid and sperm. That is why DAPI signal distribution looks different between figures 2a and 2b. All together, the results showed that prepubertal sample (s) used in our study were completely immature. We appreciate your comment and we added a statement to the figure legend to clarify the main message of this figure.

2, In Figure 6, the author did not explain the observations that the 46XY HMOs show increased death in the LIVE/DEAD staining compared to that of Klinefelter group at the same stage. Can the author provide an explanation for this observation?

Answer:

The viability of 46 XY and 47 XXY HTO did not significantly vary in terms of LIVE/DEAD staining nor ATP production. Perceived differences may be due to size variability, outlier perception of non-viable cells over a general viable cell background, and small differences in staining-wash-image acquisition as confocal LIVE/DEAD imaging was performed with freshly harvested HTO. Thanks for your comment, we replaced the image of day 23 Live/Dead staining of 46XY HTO for another technical replicant that might betterillustrate these results.

3, In Figure 6, the representative H&E staining image of the 46XY HTO on day 9 appears different from the others. If this data was collected using different methods, the author should mention this in the methods section and the figure legend.

Answer:

In Figure 6, all organoids have been harvested in the same way, fixed in the same fixative and stained with the same autostainer H&E protocol. Any differences between those images are due to the expected variability of biological replicants. Thanks for bringing it up, to make it not confusing, we replaced that image with another replicant for the same time point with the same level of Eosin staining.

4, In Figure 7, the author did not specify which group of data was used in panels A and B. For comparison, a negative control is also missing.

Answer:

The data shown in figure 7 express the evolution of HTO size and ATP production along culture. As we mention, we didn’t see any difference between 46 XY and 47 XXY organoids data but we did see differences between day 2 and day 9. To emphasize that point we pooled together the data from 46 XY and 47 XXY organoids. Thanks for this important point, we added a statement to the figure legend to cover this point.

In terms of negative control for ATP production (panel B), the luminescence signal of negative control (HTO culture medium without adding ATP kit reagents) has been subtracted as background.  A  statement has been added to the figure legend.

5, In Figure 9a, the group labels are missing. Please add the appropriate labels  to clarify the data.

Answer:

Thank you very much for finding this missing! we updated the legend labels accordingly (black: Day 2 and Gray: Day 23).

6, In Figure 10, the resolution of the images is too low. It would be better to use higher magnification and resolution images to clarify the results.

Answer:

The images in Figure 10 were acquired using fluorescence filter microscope Zeiss Axiophot microscope and image acquisition software Applied Spectral Imaging Software with the assistance of our molecular biology clinical core lab. Unfortunately, that is the best resolution we could get and we humbly believe it is on par with imaging technology currently for clinical application. We tried to show the inserts of important areas to make the signals clear.  Proving larger size of the whole image in supplementary data could be a solution if editor agrees. Therefore we will provide this Figure in a PDF/TIFF format in addition to the integrated figure in Microsoft World format.

**We will upload  a version with the mentioned edits highlighted for easier review and then final clean version will be submitted for publication

Reviewer 2 Report

Comments and Suggestions for Authors

Guillermo et al have extended their previous research where the group created HTOs from cells isolated from mature testicular tissue. The results presented here are remarkable and will pave the way for infertility treatment of patients who went to gonadotoxic treatment during childhood as well as KS patients. I have a few comments.

1. All Graphs should be presented in the dot plot method.

2. PAS staining is preferable to H&E as PAS will show clear fibrosis and a much clearer picture of spermatogenesis stages that can not be assessed through H&E. 

3. In figure 3, there is no loading control

4. In Figure 6, there are dead cells in the prepubertal group while no dead cells in the KS HTOs group. Generally, live cells go apoptosis. Is apoptosis not happening in KS HTOs? 

5.  Figure 9a, no clear depiction of the black and off-white bar.

6. Staining of c-kit would give hint that cells are going in differentiated stage.

Author Response

Reviewer 2:

Guillermo et al have extended their previous research where the group created HTOs from cells isolated from mature testicular tissue. The results presented here are remarkable and will pave the way for infertility treatment of patients who went to gonadotoxic treatment during childhood as well as KS patients. I have a few comments.

  1. All Graphs should be presented in the dot plot method.

Answer:

We believe every Graph was constructed in a format that illustrates the significance and meaning of the presented data. Some of them used dot plot method and others don’t. If there is a particular graph that doesn’t appear to transmit its intended message we will be happy try to find a better method to show.

  1. PAS staining is preferable to H&E as PAS will show clear fibrosis and a much clearer picture of spermatogenesis stages that can not be assessed through H&E. 

Answer:

For this study we have used H&E staining as it is the standard clinical histology standardized staining and the staining method we had more experience with. The reviewers make a nice suggestion about PAS staining. We are not sure if PAS staining will better assess spermatogenesis level in our HTO system but we will definitely consider it for future works.

  1. In figure 3, there is no loading control

Answer:

All primers were previously validated and tested. We used both genomic DNA and water as negative controls in our experiments with no bands observed, data not shown in the figure. A statement has been added the figure legend.

  1. In Figure 6, there are dead cells in the prepubertal group while no dead cells in the KS HTOs group. Generally, live cells go apoptosis. Is apoptosis not happening in KS HTOs? 

Answer:

The viability of 46 XY and 47 XXY HTO did not significantly vary in terms of LIVE/DEAD staining nor ATP production. Perceived differences may be due to size variability, outlier perception of non-viable cells over a general viable cell background, and small differences in staining-wash-image acquisition as confocal LIVE/DEAD imaging was performed with freshly harvested HTO. Thanks for your comment, we replaced the image of day 23 Live/Dead staining of 46XY HTO for another technical replicant that might better illustrate these results.

  1. Figure 9a, no clear depiction of the black and off-white bar.

Answer:

Thank you very much for finding this missing! we updated the legend labels accordingly (black: Day 2 and Gray: Day 23).

  1. Staining of c-kit would give hint that cells are going in differentiated stage.

Answer:

The reviewer kindly makes a great point about c-kit staining to assess differentiated stage. For this study we choose PRM1 and Acrosin as differentiation markers as we had validated them in our previous work. We confirm that data with haploid cells imaging through FISH. However, future studies will better characterize the differentiation process recreated in vitro through our system and c-kit will be one of the markers we will consider.

*We will upload a highlighted version with the recent edits for easier review. Then a clean version will be submitted for publication. 
